# Differentiating iron-loading anemias using a newly developed and analytically validated ELISA for human serum erythroferrone

Laura Diepeveen[1]*, Rian Roelofs[1], Nicolai Grebenchtchikov[1], Rachel van Swelm[1], Leon Kautz[2], Dorine Swinkels[1]

1 Translational Metabolic Laboratory, Department of Laboratory Medicine, Radboud University Medical Center, Nijmegen, The Netherlands, 2 Institut de Recherche en Santé Digestive (IRSD), Université de Toulouse, INSERM U1220, Institut National de la Recherche Agronomique (INRA) U1416, École Nationale Vétérinaire de Toulouse (ENVT), Université Paul Sabatier (UPS), Toulouse, France

* laura.diepeveen@radboudumc.nl

**Data Availability Statement:** All relevant data are within the paper and its Supporting information files.

## Abstract

Erythroferrone (ERFE), the erythroid regulator of iron metabolism, inhibits hepcidin to increase iron availability for erythropoiesis. ERFE plays a pathological role during ineffective erythropoiesis as occurs in X-linked sideroblastic anemia (XLSA) and β-thalassemia. Its measurement might serve as an indicator of severity for these diseases. However, for reliable quantification of ERFE analytical characterization is indispensable to determine the assay's limitations and define proper methodology. We developed a sandwich ELISA for human serum ERFE using polyclonal antibodies and report its extensive analytical validation. This new assay showed, for the first time, the differentiation of XLSA and β-thalassemia major patients from healthy controls (p = 0.03) and from each other (p<0.01), showing the assay provides biological plausible results. Despite poor dilution linearity, parallelism and recovery in patient serum matrix, which indicated presence of a matrix effect and/or different immunoreactivity of the antibodies to the recombinant standard and the endogenous analyte, our assay correlated well with two other existing ERFE ELISAs (both $R^2$ = 0.83). Nevertheless, employment of one optimal dilution of all serum samples is warranted to obtain reliable results. When adequately performed, the assay can be used to further unravel the human erythropoiesis-hepcidin-iron axis in various disorders and assess the added diagnostic value of ERFE.

## Introduction

In 2014, the hormone erythroferrone (ERFE) was discovered as erythroid regulator of iron metabolism [1]. Failing oxygen levels in tissues, for example during hemorrhage or hypoxia [2], result in renal synthesis of erythropoietin (EPO) [3], which, in turn, promotes expression of ERFE by erythroblasts via the JAK/STAT5 pathway [1, 4, 5]. By suppressing the key regulator of iron metabolism hepcidin, ERFE promotes duodenal iron absorption and iron mobilization from stores to meet the increased iron demand of developing erythrocytes [2, 4, 6].

**Funding:** This research was partly funded by the Noyons stipendium of the E.C. Noyons foundation. The funders had no role in study design, data collection and analysis, decision to publish, or preparation of the manuscript.

**Competing interests:** The authors have declared that no competing interests exist.

Besides its physiological role, ERFE is thought to play a crucial role in the pathophysiology of iron-loading anemias resulting from ineffective erythropoiesis, such as β-thalassemia syndromes and X-linked sideroblastic anemia (XLSA) [7].

Hemoglobin in erythrocytes contains α- and β-globin chains. Patients with β-thalassemia syndromes have gene defects leading to no or a reduced expression of β-globin, which results in an excess of free α-globin chains in erythroblasts [8, 9]. During differentiation into erythrocytes, these chains can aggregate causing formation of reactive oxygen species and damage to the cell membrane, provoking cellular apoptosis and anemia [10]. Multiple subtypes of the pathology are known, either β-thalassemia minor, intermediate or major, which depends on the amount of functional of β-globin chains left and therefore directly correlates with the severity of the disease [8].

XLSA, on the other hand, is caused by genetic defects in the erythroid-specific isoform of the aminolevulinate acid synthase (*ALAS2*) gene, which catalyzes the first and rate limiting step of heme biosynthesis in the mitochondria of erythroid precursor cells [11–13]. This results in heme and, consequently, hemoglobin deficiency, since 85% of heme is produced in the erythroblasts [14, 15]. Deposition of non-heme iron in the mitochondria of these erythroid precursors causes formation of so-called ring sideroblasts [15]. *ALAS2* is located on the X-chromosome and is recessively expressed, leaving mainly men affected.

Production of dysfunctional erythroblasts in these pathologies, that do not successfully differentiate into mature erythrocytes, leads to anemia and increased synthesis of EPO [16]. Hence, ERFE concentrations further increase in order to match iron supply to the erythropoietic demand, while hepcidin remains suppressed. However, in these pathologies, erythrocyte production cannot be restored leaving the excess amounts of iron unused resulting in iron overload [7, 16]. Patients with β-thalassemia major are transfusion-dependent due to the severity of ineffective erythropoiesis, in turn aggravating the iron overload [9].

General protocols are already in place to establish the diagnosis of diseases of ineffective erythropoiesis. Yet, considering the role of ERFE in diseases of ineffective erythropoiesis, quantification of systemic ERFE levels might provide clinicians with supporting information regarding disease severity. Moreover, ERFE measurement could be clinically relevant in additional pathologies, since serum ERFE levels were found to predict mortality and cardiovascular events in both chronic kidney disease and hemodialysis patients [17]. In addition, genetic *ERFE* variants were recently discovered that contribute to hepcidin suppression and subsequent body iron overload in some patients with myelodysplastic syndromes and congenital dyserythropoietic anemia [18, 19]. It can be hypothesized that individuals with gain-of-function mutations in the *ERFE* gene might recover faster from blood loss. This would especially be interesting in the blood donor population to personalize donation intervals, also since identification of *ERFE* variants that delay recovery of iron stores could contribute to prevent iron deficiency. In order to correctly assess ERFE levels in pathologies and after blood donation, appropriate analytical validated quantification of ERFE must be ensured. This will also contribute to correctly evaluate the success of recently suggested anti-ERFE therapies for pathologies with aberrantly high ERFE levels, such as iron-loading anemias [20, 21].

Currently, only one validated in-house monoclonal ERFE enzyme-linked immunosorbent assay (ELISA) has been published [22], and several commercial kits are on the market. However, none is yet routinely available [23] and solely used for research. Our experience in developing assays for the iron-related measurements [24–29] has taught us the importance of validation and standardization before using an assay in clinical practice [30–32]. Full analytical characterization of an assay ensures correct utilization (fit-for-purpose) of the method and appropriate interpretation of the data, as it reveals the assay's limitations and therefore defines proper methodology to produce reliable results for diagnostic and research purposes [33]. To

this end, we aimed to develop a robust, analytically characterized and eventually standardized in-house ERFE ELISA, which will guarantee reproducible and consistent ERFE results. Here, we describe the analytical validation of a polyclonal antibodies-based sandwich ELISA to measure ERFE concentration in human serum. The assay was biologically validated by measuring ERFE concentrations in healthy controls and in patients with iron loading anemias β-thalassemia major and XLSA and was compared with two existing ERFE assays.

## Materials and methods

### Polyclonal anti-hERFE antibodies

Catching and tagging antibodies were generated by immunizing two chickens and two rabbits, respectively, with 10 μg of recombinant Flag-tagged human ERFE (hERFE) immunogen per injection by Davids Biotechnology (Regensburg, Germany) and purified using affinity chromatography (details are described in S1 Methods). All animal experiments related to the generation of the antibodies were outsourced to Davids Biotechnology.

### Samples

**Patients and controls.** Serum samples of 7 patients with β-thalassemia major (4 men, 3 women; age 20–49 yrs) and 7 patients diagnosed with XLSA (7 men; age 16–72 yrs) were obtained from the Iron Biobank that is part of the centralized Radboud Biobank of the Radboudumc (Nijmegen, the Netherlands) [34]. In addition, blood samples were collected from 15 healthy volunteers (7 men, 8 women; age 20–56 yrs). Additional laboratory measurements and the detailed sample collection are described in S1 Methods. Lastly, we collected leftover serum and heparin plasma samples from phlebotomy of 5 XLSA patients. Each serum-plasma pair was collected at the same time.

**Ethical approval.** The collection and usage of both the patient samples and the healthy volunteer samples were approved by the Ethics Committee (Arnhem-Nijmegen, the Netherlands) and the Board of Directors of the Radboudumc (Nijmegen, The Netherlands) and this study has been conducted in accordance with the Declaration of Helsinki. All collected samples were coded and informed consents were signed prior to inclusion in the Radboud Biobank and blood collection.

Leftover patient material used in this study was fully anonymized upon collection and therefore conform with the code for proper secondary use of human tissue in the Netherlands.

### ELISA method

96-well plates were coated with chicken anti-hERFE. After washing and blocking, the hERFE standard (10 ng/μL) was serial diluted between 4 and 0.125 ng/mL and incubated overnight together with the diluted serum samples, all in duplicate. The plate was washed and incubated with rabbit anti-hERFE antibodies for 1 hour, followed by a 1 hour incubation with anti-rabbit IgG-HRP antibody. The signal was developed with TMB One and stopped with 0.2M $H_2SO_4$ after 20 minutes, after which optical density (OD) was measured at 450 nm. Details on the protocol, ELISA format and equipment and reagents used are described in S1 Methods.

### Analytical validation of ERFE assay

The limit of blank (LOB) was determined by calculating the ERFE concentration corresponding to the mean OD of the dilution buffer (83 replicates). The limit of detection (LOD) was calculated as LOB + 1.645 * standard deviation of blanks ($sd_B$) [35].

The precision of the assay was calculated using a sample with a low, middle and high concentration of ERFE. Both the intra-assay coefficient of variation (CV,%) (15 consecutive replicates in a single assay for the middle and high samples, 12 for the low sample) and the inter-assay CV% (5 replicates carried out on 3 separate days for the middle and high samples, 4 replicates for the low sample) were determined.

Dilution linearity was assessed to demonstrate that a sample with a spiked concentration above the upper limit of quantification (ULOQ) can be diluted to a concentration within the range of the standard curve and still give a reliable result, investigating the effect of diluting the analyte in dilution buffer. To this end, two control samples (0.25 and 0.04 ng/mL) were spiked with hERFE to 400 ng/mL. The diluted samples were measured in one run and recovery was calculated by ((measured ERFE / expected ERFE) * 100%).

Parallelism, which is conceptually similar to dilution linearity, was assessed by a serial dilution (5, 10, 15, 20, 40 fold) of patient samples with high endogenous ERFE concentrations (n = 9), all measured in one run. Using samples with high endogenous levels of the analyte, potential sample matrix effects are investigated and the binding characteristic of the endogenous analyte to the antibodies can be compared to the binding characteristics of the standard. By studying different dilution factors as reference value to calculate recovery, the minimum required dilution of the assay was determined. The minimum required dilution represents the smallest dilution giving a constant recovery after further dilution [36].

Recovery was studied to investigate whether the dose-response relationship of the analyte differs in standard diluent or sample matrix, by spiking two control samples (1.21 and 0.46 ng/mL) and a dilution buffer sample with a mix composed of samples with low, medium-low, medium-high and high hERFE concentration. All were measured within the same run and recovery was calculated by ((measured ERFE spiked—measured ERFE neat)/theoretical concentration ERFE)*100%).

Differences in ERFE measurement for serum and heparin plasma were studied in XLSA patient sample pairs (n = 5) and measured in one run.

Sample stability was assessed by storing aliquots of two patient samples up to 48 hours at room temperature, -4˚C and for 1 month at -20˚C. Additionally, both samples were freeze-thawed up to 6 times. ERFE concentrations were compared to baseline measurements.

## Statistical analysis

All statistical analyses were performed with Graphpad Prism (version 5.03). Standard curves were approximated using four parameter logistic dose-response curves as $Y = D+(A-D)/(1+(X/C)^B)$ in which X stands for the analyte concentration and Y for absorbance value. Coefficient A represents the bottom of the curve, B represents Hill's slope, C represents the inflection point of the curve and D represents the top of the curve. Interpolation is performed with log transformed concentrations. Additionally, independent variables were compared using a t-test (two-groups) or ANOVA with a Bonferroni post-test (more than two groups). The difference between serum and plasma ERFE measurements were compared using a paired one-tailed t-test. Agreement between assays was studied using linear regression analysis. Two-sided P-values below 0.05 were considered to be statistically significant.

## Results

### Assay characteristics

LOB and LOD in the assay were found to be 0.036 and 0.064 ng/mL, respectively. To assess the precision of the assay, intra- and inter-assay CV were determined for low, middle and high ERFE levels (Table 1). We observed reduced precision for lower ERFE concentrations and an

**Table 1. Three-level precision.**

| ERFE level | Concentration (ng/mL) (mean ± SD) | Coefficient of variation (CV%) | |
| --- | --- | --- | --- |
| | | Intra-assay | Inter-assay |
| Low | 2.4 ± 0.5 | 20.4 | 24.3 |
| Middle | 3.8 ± 0.5 | 12.8 | 17.0 |
| High | 6.0 ± 0.4 | 6.6 | 17.2 |
| | | Overall: 13.3 | Overall: 19.5 |

Precision was studied in three samples with either a low, middle or high ERFE concentration. Intra-assay CV% was determined by either 15 replicate measurements for the middle and high samples or 12 for the low sample. Inter-assay CV% was determined by either 5 replicates carried out on 3 separate days for the middle and high samples or 4 for the low sample.

intra-assay CV <10–15% for the middle and high sample. Overall intra- and inter-assay CVs were 13.3% and 19.5%, respectively. The intra-assay was smaller than the inter-assay CV at all levels, indicating that the variation between runs is higher than on the same run.

Next, we aimed to investigate how samples must be diluted in the assay to ensure reliable results by studying dilution linearity and parallelism.

Dilution linearity was investigated by spiking two serum samples containing low ERFE concentrations with hERFE and diluting them to a concentration within the range of the calibration curve. Interestingly, a dilution factor of 5 appeared insufficient to dilute the samples below the ULOQ (Table 2). For the other dilutions, recovery ranged between 61–114%, while standard criteria accept 80–120% [37]. These results might indicate a nonlinear dose-response relationship of the analyte in dilution buffer and necessitates using one fixed dilution for all samples to allow their comparison to be reliable.

By studying parallelism, we aimed to find the optimal dilution resulting in a constant recovery after further dilution by serially diluting serum samples with high endogenous ERFE concentrations. All dilution factors (5, 10, 15, 20 and 40) were tested and the minimum required dilution factor was found to be 15, as this dilution showed a recovery of which the subsequent dilution (1:20) remained most constant. Using 1:15 as reference value, the mean recovery of all nine samples with a 1:20 dilution was found to be 116% with a range of 75–162% (Table 3). This recovery range showed to be the narrowest compared to the results of other dilution factors as reference value (S1 Table), closest to the standard accepted criteria of 75–125% [38]. However, as the range shows, not all samples showed a recovery within these criteria (Table 3,

**Table 2. Dilution linearity.**

| Dilution factor | Expected [ERFE] (ng/mL) | Sample 1 | | Sample 2 | |
| --- | --- | --- | --- | --- | --- |
| | | Measured [ERFE] (ng/mL) | Recovery(%) | Measured [ERFE] (ng/mL) | Recovery (%) |
| 0 | 400 | >> | | >> | |
| 5 | 80 | >> | | >> | |
| 25 | 16 | 16.46 | 103 | 11.81 | 74 |
| 125 | 3.20 | 2.22 | 70 | 1.96 | 61 |
| 625 | 0.64 | 0.52 | 82 | 0.52 | 81 |
| 3125 | 0.13 | 0.15 | 114 | 0.14 | 113 |

Dilution linearity assessment of ERFE assay in human serum. Two low ERFE containing serum samples were spiked with hERFE to 400 ng/ml and serially diluted. Recovery was calculated as (measured ERFE / expected ERFE) *100%. Measured ERFE concentrations were not dilution-adjusted. >> represents ERFE concentrations above ULOQ.

**Table 3. Parallelism using the concentration at 1:15 as a reference value.**

| Dilution factor | Sample 1 Conc. (ng/mL) | Sample 1 Recov. (%) | Sample 2 Conc. (ng/mL) | Sample 2 Recov. (%) | Sample 3 Conc. (ng/mL) | Sample 3 Recov. (%) | Sample 4 Conc. (ng/mL) | Sample 4 Recov. (%) | Sample 5 Conc. (ng/mL) | Sample 5 Recov. (%) | Sample 6 Conc. (ng/mL) | Sample 6 Recov. (%) | Sample 7 Conc. (ng/mL) | Sample 7 Recov. (%) | Sample 8 Conc. (ng/mL) | Sample 8 Recov. (%) | Sample 9 Conc. (ng/mL) | Sample 9 Recov. (%) |
|---|---|---|---|---|---|---|---|---|---|---|---|---|---|---|---|---|---|---|
| 5 | 2.59 | 126 | 2.45 | 97 | 1.10 | 75 | 4.64 | 75 | 3.62 | 75 | 2.64 | 73 | 3.67 | 50 | 1.22 | 41 | 3.28 | 176 |
| 10 | 1.93 | 94 | 2.93 | 116 | 2.32 | 159 | 5.40 | 87 | 1.59 | 33 | 3.84 | 107 | 4.23 | 58 | 1.43 | 48 | 3.71 | 199 |
| **15** | **2.05** | **100** | **2.53** | **100** | **1.46** | **100** | **6.20** | **100** | **4.84** | **100** | **3.60** | **100** | **7.29** | **100** | **2.97** | **100** | **1.86** | **100** |
| 20 | 2.20 | 107 | 1.88 | 75 | 1.37 | 94 | 8.78 | 142 | 5.51 | 114 | 5.84 | 162 | 8.06 | 111 | 3.32 | 112 | 2.28 | 123 |
| 40 | 3.47 | 169 | 3.48 | 138 | 4.03 | 276 | 8.18 | 132 | 1.82 | 38 | 4.84 | 134 | 6.04 | 83 | << | << | 0.66 | 35 |

Parallelism assessment of ERFE assay in human serum. Nine patient samples with high endogenous ERFE concentrations were serially diluted. Recovery was calculated using the ERFE concentration determined at dilution 1:15 as a reference value. Concentrations were dilution-adjusted. << represent ERFE concentrations below LLOQ.

**Table 4. Difference in ERFE concentration between serum and (heparin) plasma.**

| Patient ID | ERFE concentration in serum (ng/mL) | ERFE concentration in plasma (ng/mL) |
|---|---|---|
| 1 | 2.82 | 3.09 |
| 2 | 4.03 | 5.00 |
| 3 | 2.25 | 3.14 |
| 4 | 1.30 | 1.97 |
| 5 | 3.70 | 4.84 |

ERFE was measured in serum and heparin plasma samples of the same blood draw of 5 XLSA patients. ERFE concentrations in plasma were significantly higher compared to those measured in serum (paired one-tailed t-test, p = 0.0031).

sample 4 and 6, 142% and 162%, respectively), suggesting possible interferences from sample matrix and/or differences in immunoreactivity of the antibodies between the endogenous analyte and the standard. Yet, using one fixed dilution will circumvent discrepancies in ERFE measurements between samples and strengthen the reproducibility of the results. Since recovery of the 1:40 dilution with 1:20 as reference value did not show to be constant (S1 Table; mean 119%, range 29–295%), the optimal dilution factor must be between 15 and 20. To this end, we chose 1:18 as standard dilution of the ERFE assay to use in further measurements.

Recovery of ERFE was studied by spiking two low ERFE containing control samples with hERFE concentrations of 0.88, 10.00, 40.00 and 72.45 ng/mL. This resulted in a recovery of 63 ±16% (mean±SD), 52%±11%, 44±5% and 37±0%, respectively, demonstrating that recovery in serum samples decreased with increasing spiking concentrations. Spiking of hERFE in dilution buffer resulted in higher recovery (67%, 74%, 103% and 92%, respectively), indicating a different dose-response of the analyte in standard diluent and sample matrix, thus confirming the existence of a matrix effect.

Differences in ERFE measurement between serum and heparin plasma samples of 5 XLSA patients were studied to explore whether both sample materials provide similar results. In all samples, ERFE concentrations measured in heparin plasma samples were significantly higher (Table 4, p = 0.0031).

Stability experiments showed that ERFE concentrations measured in serum samples remained stable for 4 freeze-thaw steps and during at least a 24-hour storage at RT, 48 hours at 4˚C and 1 month at -20˚C.

## Comparison with other ERFE assays

As an extension of the analytical validation, we compared our ELISA to a commercial ERFE kit [39] in 20 serum samples and to an in-house assay [22] in 23 serum samples. Samples were selected to cover a broad range of ERFE concentrations, including both patient and control samples. Linear regression showed good correlation with both the commercial kit ($R^2$ = 0.8258, p<0.0001) and the in-house assay ($R^2$ = 0.8332, p<0.001) (Fig 1). Equations of the regression lines were found to be Y = 12.62X-13.25 and Y = 18.92X-16.92, respectively, indicating large differences in absolute values between the assays despite the good correlation, which hampers the comparability of the ERFE measurements between methods.

## ERFE concentrations of XLSA and β-thalassemia patients

To biologically validate the assay, ERFE concentrations were studied in serum samples of 7 XLSA, 7 β-thalassemia major patients and 15 healthy controls. The summary of descriptive and hematological characteristics of both patients and controls are given in Table 5 (extended

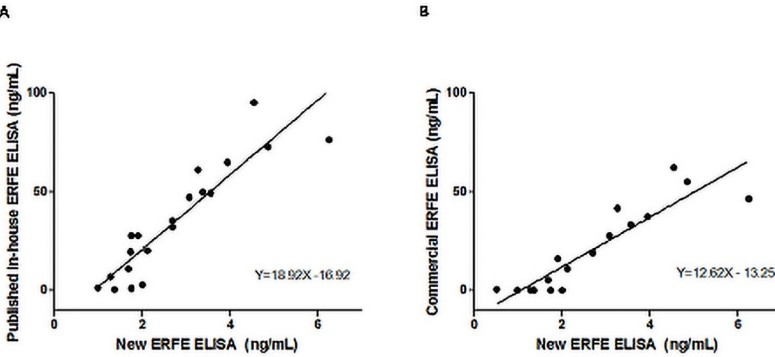

**Fig 1. Comparison of our new ERFE ELISA with a published in-house assay (A) and commercial kit (B) using linear regression analysis.** (A), n = 23 serum samples or (B) n = 20 serum samples. Statistical analysis was performed using linear regression analysis. (A: $R^2 = 0.8332$, $p<0.001$; B: $R^2 = 0.8258$, $p<0.0001$).

information in S2 Table). We demonstrated that ERFE levels of healthy controls significantly differ from those found in XLSA and β-thalassemia major patients (Fig 2, overall p = 0.0264) and also that ERFE concentrations of both iron loading anemias significantly differ from each other (p<0.05), thus confirming the biological plausibility of the assay.

## Discussion

This study describes the development and subsequent analytical and biological validation of a new sandwich ELISA for human ERFE. We presented the analytical assay characteristics and limitations, which are important to comprehend, since these define how the assay must be used to ensure correct quantification of the analyte [33]. The assay showed good correlation with two previously validated assays, and proves to differentiate ERFE levels of iron loading

**Table 5. Summary of patient and control characteristics and hematological parameters.**

|  | Hb (mM)[a] | MCV (fl) | Iron (µM) | TIBC (µM) | TSAT (%) | Ferritin (µg/L) | CRP (mg/L) | EPO (U/L) | Hepcidin (nM)[b] | Hepcidin/ferritin ratio (pmol/µg) | ERFE (ng/mL) |
|---|---|---|---|---|---|---|---|---|---|---|---|
| **Diagnosis** | | | | | | *Median (range)* | | | | | |
| β-thal. M | 5.5 (4.8–5.8) | 79.0 (67–83) | 36 (26–46) | 43 (32–49) | 94 (78–105) | 721 (563–5376) | 1.0 (<1–1) | 56.3 (39.8–199) | 3.0 (<0.5–7.5) | 1.56 (0.89–6.77) | 3.95 (2.70–6.26) |
| XLSA | 7.7 (6.8–8.0) | 73.5 (64–75) | 28 (19–38) | 58 (43–66) | 42 (40–88) | 226 (58–573) | 1.5 (<1–2) | 10.2 (7.4–22.8) | 3.6 (<0.5–8.9) | 15.53 (0.65–56.90) | 2.13 (1.74–3.57) |
| Controls | ND | ND | 19 (5–34) | 69 (50–92) | 28 (7–52) | 86 (11–385) | 1.0 (<1–2) | 8.0 (4.2–22.6) | 2.3 (<0.5–12.1) | 19.38 (11.90–55.56) | 1.44 (<1.16–2.01) |
| *Normal range values* [c] | > 7.5 | 80–95 | 10–30 | 45–78 | 16–45 | 20–300 | < 5.0 | 4–29 | < 0.5–15.5 | 3.1–176.4 | ND |

[a] g/L = 16.1 mM.

[b] Standardized hepcidin value.

[c] Normal range values of men and (pre-menopausal) women were combined and provided by the executive laboratories of the measurements.

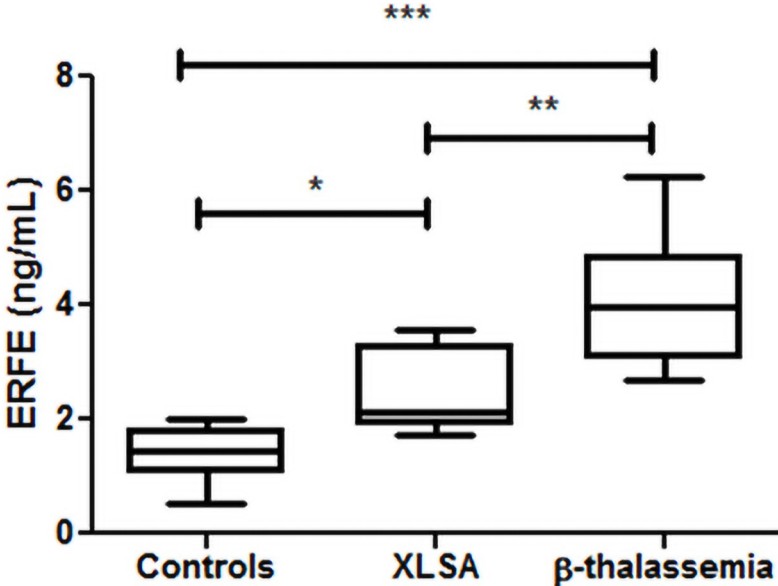

**Fig 2. Serum ERFE concentrations in two iron loading anemias.** ERFE measurements of healthy controls, X-linked sideroblastic anemia patients and β-thalassemia major patients using the newly developed ERFE assay. Boxes show median and second and third quartiles. Whiskers extend to minimum and maximum values. Statistical analysis was performed using ANOVA with Bonferoni's post-test. * p < 0.05, ** p < 0.001, *** p < 0.0001.

anemias β-thalassemia major and XLSA from healthy controls. More importantly, we demonstrated that these pathologies can be differentiated from each other based on ERFE measurement.

Currently, correctly validated human ERFE ELISAs are scarce. A previous study reported poor analytical performance of two commercially available ERFE ELISA kits [40] and also our study shows the importance of extensive analytical validation. The assay was validated with serum samples. ERFE concentrations were found to be higher in heparin plasma samples compared to serum samples, corroborating previous findings [40], demonstrating that ERFE results obtained for serum and heparin plasma are not interchangeable. Furthermore, the assay showed poor dilution linearity, parallelism and recovery in sample serum matrix. These findings are not uncommon for assays based on immunobinding due to matrix interference in the antibody-antigen binding [36]. Yet, this does not negate the use of the assay, but is rather critical knowledge for correct use of the assay and data interpretation.

Poor dilution linearity indicates inflexibility of the assay to correctly quantify ERFE concentrations at different dilutions. On the other hand, poor parallelism demonstrates that binding of the antibodies to the recombinant standard differs from that of the endogenous analyte in serum matrix [41]. A standard of recombinant material is used, since the endogenous analyte is not available, which might explain these results. Hence, suggested multimerization of endogenous ERFE or possible existence of isoforms, which is characteristic for the family of proteins to which ERFE belongs [20, 22, 42], might also contribute to these findings. These posttranslational processes are currently poorly understood, which complicates predicting subsequent effects on ERFE quantification by immunoassays. Combined, poor dilution linearity and parallelism necessitate working with one optimal dilution. Furthermore, poor parallelism might indicate presence of a matrix effect [43, 44]. Matrix effects in an ELISA reflect factors that affect the antigen-antibody binding, such as presence of heterophilic antibodies, endogenous

sample components as enzymes and immunoglobulins or exogenous substances as polymers and detergents [45]. Although we tried to avoid the influence of heterophilic antibodies by using a 'nonsense format', our results clearly show presence of a matrix effect, especially since poor recovery was found from ERFE spiked in sample matrix and improved with ERFE spiked in dilution buffer. Assessing a minimum required dilution, and employing it as one standard dilution at which the matrix effect is alleviated, allows the analytically proper use of the assay [36, 44].

The notion that the assay produces reliable results, despite its limitations, is strengthened by the good correlation found between our assay and both the validated commercial kit ($R^2$ = 0.8258, p<0.0001) and in-house assay ($R^2$ = 0.8332, p<0.001). Nevertheless, the comparison revealed large difference in absolute ERFE concentrations between the different assays, indicating the lack of harmonization and standardization. Only global implementation of a secondary reference material can solve the differences in absolute values, which will enable establishment of reference values and worldwide comparison of ERFE results in both diagnostics and research. With a secondary reference material, which is a commutable matrix-based working standard, harmonization can be achieved leading to equivalent ERFE results [46]. Obtaining a true value is achieved by standardization, for which the secondary reference material is calibrated by a reference method with a well characterized primary reference material for ERFE of certified purity.

The biological plausibility of the newly developed ERFE assay is supported by its capability to differentiate β-thalassemia major and XLSA patients from healthy controls and, interestingly, from each other. ERFE concentrations were found to be significantly higher in β-thalassemia major patients compared to XLSA patients. This can be attributed to increased severity of ineffective erythropoiesis in β-thalassemia major patients, that results in lower Hb levels and a more frequent need of blood transfusions [7, 23]. Therefore, quantification of ERFE might serve as a marker for severity of ineffective erythropoiesis and may eventually be a used in diagnostics of iron loading anemias and to predict the risk of body iron overload in these diseases.

Altogether, we demonstrate that the newly developed assay is well-suited to improve insights in the erythropoiesis-hepcidin-iron axis by disorder comparison studies and might also be valuable to assess trends in ERFE after for instance blood loss or iron supplementation. In recent years, the role of ERFE has been studied in a variety of research fields, including energy metabolism [47] and chronic kidney disease [48]. Furthermore, the molecular mechanisms by which ERFE is produced and exerts its function are being unraveled [6, 42, 49–51]. ERFE measurement allows us to study the above mentioned axis in hematological patients other than those characterized by ineffective erythropoiesis, such as anemia of chronic disease and in phenotypes causing hypoxia such as chronic obstructive pulmonary disease, unraveling pathophysiological mechanisms. Additionally, its measurement could help detecting and studying stimulation of erythropoiesis by doping [52] and altitude training [53]. However, increased (patho)physiological insights in research studies and as relevant outcome measures in clinical trials, not only depends on the availability of accurate and precise assays, but also on their correct application, interpretation of its measurement and quality of the produced data [38]. Therefore, we stress the importance of elaborate analytical validation prior to the use of any newly developed assay.

In conclusion, we developed a new in-house ERFE ELISA, which we validated to ensure correct utilization of the assay and interpretation of the data for diagnostic purposes and research studies. As a next step, we aim to better understand the characteristic features of ERFE as a measurand, as well as the matrix effects on the antibody-antigen binding. This will further improve the analytical performance of assays, which lead to analytically robust

methods. Developing secondary and primary reference materials would allow harmonization/ standardization of ERFE assays, which will contribute to the success of the biomarker by enabling comparison of absolute values obtained by different assays.

## Supporting information

**S1 Methods. Supplemental methods.**
(DOCX)

**S1 Table. Extended parallelism studies using different concentrations as reference value.**
Nine patient samples with high endogenous ERFE concentrations were serially diluted. Recovery was calculated using the ERFE concentration determined at dilution 1:5, 1:10 or 1:20 as a reference value. Mean recovery of the subsequent dilution (1:10, 1:15 or 1:40) was 117% (range 44–211%), 133% (range 50–305%) or 119% (range 29–295%), respectively. $\ll$ represent ERFE concentrations below LLOQ.
(DOCX)

**S2 Table. Patient and control characteristics and hematological parameters.** * Treatment at the time of sample collection. ** g/L = 16.1 mM. *** Standardized hepcidin value.
(DOCX)

## Acknowledgments

We would like to thank Tomas Ganz and his lab for measuring our samples in order to compare the results of their assay with ours. In addition, we thank Siem Klaver and the clinicians of the Radboudumc expertise center for iron disorders; Drs Alexander Rennings, Saskia Schols, Britta Laros, Paul Brons and Eva Rettenbacher for their contribution to the iron biobank that made this study possible and all patients and volunteers for participation.

## Author Contributions

**Conceptualization:** Dorine Swinkels.

**Data curation:** Rian Roelofs.

**Formal analysis:** Laura Diepeveen, Rian Roelofs.

**Funding acquisition:** Rachel van Swelm, Dorine Swinkels.

**Investigation:** Laura Diepeveen, Rian Roelofs, Nicolai Grebenchtchikov.

**Methodology:** Rian Roelofs, Nicolai Grebenchtchikov, Leon Kautz, Dorine Swinkels.

**Project administration:** Laura Diepeveen, Rian Roelofs.

**Resources:** Laura Diepeveen, Rian Roelofs, Nicolai Grebenchtchikov, Leon Kautz, Dorine Swinkels.

**Supervision:** Rachel van Swelm, Dorine Swinkels.

**Validation:** Laura Diepeveen, Rian Roelofs, Nicolai Grebenchtchikov.

**Visualization:** Laura Diepeveen.

**Writing – original draft:** Laura Diepeveen.

**Writing – review & editing:** Rian Roelofs, Rachel van Swelm, Leon Kautz, Dorine Swinkels.

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
