## [Decision Letter · Decision Letter 0]

8 Jun 2021

PONE-D-21-13559

Differentiating iron-loading anemias using a newly developed and analytically validated ELISA for human serum erythroferrone

PLOS ONE

Dear Dr. Diepeveen,

Thank you for submitting your manuscript to PLOS ONE. After careful consideration, we feel that it has merit but does not fully meet PLOS ONE’s publication criteria as it currently stands. Therefore, we invite you to submit a revised version of the manuscript that addresses the points raised during the review process.

Both reviewers appreciated the importance in developing and validating a new ELISA assay for ERFE. However, they also raised issues about the cohort of patients used in this work, the comparison of the new assay with another available ERFE ELISA kit, and the interpretation of some data. These issues should be addressed in a revision.

We look forward to receiving your revised manuscript.

Kind regards,

Kostas Pantopoulos, PhD

Academic Editor

PLOS ONE

Journal Requirements:

Additional Editor Comments (if provided):

Reviewers' comments:

Reviewer's Responses to Questions

**Comments to the Author**

1. Is the manuscript technically sound, and do the data support the conclusions?

Reviewer #1: Yes

Reviewer #2: Yes

2. Has the statistical analysis been performed appropriately and rigorously? 

Reviewer #1: Yes

Reviewer #2: Yes

3. Have the authors made all data underlying the findings in their manuscript fully available?

Reviewer #1: Yes

Reviewer #2: Yes

4. Is the manuscript presented in an intelligible fashion and written in standard English?

Reviewer #1: Yes

Reviewer #2: No

5. Review Comments to the Author

Reviewer #1: The manuscript describes a new ERFE Elisa assay, which is thoroughly validated for use with human plasma. The laboratory has extensive previous experience with hepcidin assays, and it is expected that the described erythroferrone assay will contribute to the ongoing studies regarding the role and mode of action of erythroferrone. The steps taken to validate of the assay should result in increased confidence in obtained data.

There are some minor points which should be addressed:

1) The authors repeatedly point out that their assay could be useful in clinical practice - as an example, they demonstrate the ability of the assay to distinguish between thalassemia major and X-linked sideroblastic anemia. In this respect, they probably should point out that the determination of plasma ERFE provides only supplementary information for the clinician. Differential diagnosis of thalassemia major vs. XLSA will always be based on overall hematologic parameters and the presence/absence of ring sideroblasts. In the case of blood donors, the main criteria will always be hemoglobin and ferritin.

2) The authors repeatedly point out the association between ERFE and ineffective erythropoiesis. However, for the testing of their assay, they use transfused thalassemia major patients. The cited reference (7) by Camaschella and Nai explicitly mentions non-transfusion-dependent thalassemia as a typical example of a disease associated with ineffective erythropoiesis. Generally, the text (for example lines 66-71) gives the impression that ERFE is the main factor influencing iron loading in thalassemia major. This is certainly not the case, given the need for repeated transfusions in thalassemia major. The need for transfusions, as well as the consequences of transfusions in thalassemia major patients should be more emphasized in the text. Did the authors consider the possibility of testing thalassemia intermedia patients?

3) Line 305: The authors probably mean "ERFE" rather than "hepcidin" (?).

4) Table 4: The difference in values for plasma and serum is interesting. Did the authors have the opportunity to check this effect with the other erythroferrone assays?

5) Table 5: The reader could benefit from adding the normal range values.

Overall, the validation of the assay appears to be competently done, and the assay itself has the potential to provide interesting data regarding the role of erythroferrone in iron metabolism.

Reviewer #2: It will be interessting to compare with human ERFE ELISA described in Ramirez Cuevas K et al, Drug Test Anal. 2020. Also, optimal samples to test are samples from volunteers injected with rhEPO. It should be added as other experiment.

6. PLOS authors have the option to publish the peer review history of their article (what does this mean?). If published, this will include your full peer review and any attached files.

Reviewer #1: No

Reviewer #2: No

---

## [Author Response · Author response to Decision Letter 0]

15 Jun 2021

Additional Journal Requirements

Authors:

The revised files were checked to meet PLOS ONE's style requirements. 

Authors:

We checked the reference list. Although no references were retracted, we noticed reference 15 and 29 were incorrect. Therefore, reference 15 was replaced and reference 29 was textually changed. 

Reviewer #1

1) The authors repeatedly point out that their assay could be useful in clinical practice - as an example, they demonstrate the ability of the assay to distinguish between thalassemia major and X-linked sideroblastic anemia. In this respect, they probably should point out that the determination of plasma ERFE provides only supplementary information for the clinician. Differential diagnosis of thalassemia major vs. XLSA will always be based on overall hematologic parameters and the presence/absence of ring sideroblasts. In the case of blood donors, the main criteria will always be hemoglobin and ferritin.

Authors:

The reviewer is correct that ERFE measurement would only provide supplementary information to clinicians regarding the diagnosis and prognosis of XLSA and b-thalassemia major. Therefore, we propose the assay can be used as indicator of severity of the phenotypes and we use the subsequent experiment, in which we aim to differentiate between the two pathologies, as biological validation of functionality of the assay. We clarified both the future utility of ERFE measurement as the goal of our experiment (line 73-76, 284, 346).

2) The authors repeatedly point out the association between ERFE and ineffective erythropoiesis. However, for the testing of their assay, they use transfused thalassemia major patients. The cited reference (7) by Camaschella and Nai explicitly mentions non-transfusion-dependent thalassemia as a typical example of a disease associated with ineffective erythropoiesis. Generally, the text (for example lines 66-71) gives the impression that ERFE is the main factor influencing iron loading in thalassemia major. This is certainly not the case, given the need for repeated transfusions in thalassemia major. The need for transfusions, as well as the consequences of transfusions in thalassemia major patients should be more emphasized in the text. Did the authors consider the possibility of testing thalassemia intermedia patients?

Authors:

We agree with the reviewer that we should provide more information on both the need as well as the consequences of transfusions in b-thalassemia major patients. We included this in the manuscript (line 71-73). Regarding the thalassemia intermedia patients, indeed it would be interesting to use this patient population. Yet, the Radboudumc biobank, that includes patient samples of the Dutch Center for Iron Disorders, could only provide us with thalassemia major patient samples. 

3) Line 305: The authors probably mean "ERFE" rather than "hepcidin" (?).

Authors:

The referee is correct. Thank you, we changed it. 

4) Table 4: The difference in values for plasma and serum is interesting. Did the authors have the opportunity to check this effect with the other erythroferrone assays?

Authors:

In this current study, we did not have the opportunity to check with the ILS kit and the other in-house assay. However, this characteristic has been tested before in a study by Appleby et al. (reference 40) who did use the ILS kit and interestingly observed the same effect. 

5) Table 5: The reader could benefit from adding the normal range values.

Authors:

We agree with the referee and added this information to Table 5. 

Reviewer #2

1) It will be interesting to compare with human ERFE ELISA described in Ramirez Cuevas K et al, Drug Test Anal. 2020. Also, optimal samples to test are samples from volunteers injected with rhEPO. It should be added as other experiment.

Authors:

Thank you for your suggestions. It would indeed be interesting to compare our results with the AdipoGen Life Sciences ERFE kit, as used by Cuevas et al. Besides this kit, more commercially ERFE ELISAs are currently increasingly available. Most of these assays are not published and data on extensive analytical validation is lacking. To increase applicability and reliability of these methods, eventually future studies should focus on harmonization or standardization of these assay, for which overall comparison of all assays is needed to ensure good analytical performance since this a prerequisite for reducing inter-assay variation. Yet, this is out of the scope of our manuscript. For the current study, we performed the comparison experiment to provide extra data on the reliability of our assay and therefore chose to solely compare our ELISA with two existing methods. 

Moreover, we agree that testing our ELISA with samples from volunteers injected with rhEPO would increase the biological plausibility of our assay and will keep this in mind for future experiments. However, since the most important message of our paper is to extensively analytically validate an assay in order to use the assay correctly, we believe one biological validation experiment is sufficient. In addition, the suggested experiment would require approval according to the Dutch Medical Research Involving Human Subjects acts, which is currently not feasible for this manuscript.

---

## [Decision Letter · Decision Letter 1]

5 Jul 2021

Differentiating iron-loading anemias using a newly developed and analytically validated ELISA for human serum erythroferrone

PONE-D-21-13559R1

Dear Dr. Diepeveen,

We’re pleased to inform you that your manuscript has been judged scientifically suitable for publication and will be formally accepted for publication once it meets all outstanding technical requirements.

Kind regards,

Kostas Pantopoulos, PhD

Academic Editor

PLOS ONE

Additional Editor Comments (optional):

Reviewers' comments:

Reviewer's Responses to Questions

**Comments to the Author**

1. If the authors have adequately addressed your comments raised in a previous round of review and you feel that this manuscript is now acceptable for publication, you may indicate that here to bypass the “Comments to the Author” section, enter your conflict of interest statement in the “Confidential to Editor” section, and submit your "Accept" recommendation.

Reviewer #1: All comments have been addressed

2. Is the manuscript technically sound, and do the data support the conclusions?

Reviewer #1: (No Response)

3. Has the statistical analysis been performed appropriately and rigorously? 

Reviewer #1: (No Response)

4. Have the authors made all data underlying the findings in their manuscript fully available?

Reviewer #1: (No Response)

5. Is the manuscript presented in an intelligible fashion and written in standard English?

Reviewer #1: (No Response)

6. Review Comments to the Author

Reviewer #1: (No Response)

7. PLOS authors have the option to publish the peer review history of their article (what does this mean?). If published, this will include your full peer review and any attached files.

Reviewer #1: No

---

## [Editor Report · Acceptance letter]

9 Jul 2021

PONE-D-21-13559R1 

Differentiating iron-loading anemias using a newly developed and analytically validated ELISA for human serum erythroferrone 

Dear Dr. Diepeveen:

I'm pleased to inform you that your manuscript has been deemed suitable for publication in PLOS ONE. Congratulations! Your manuscript is now with our production department. 

Kind regards, 

on behalf of

Dr. Kostas Pantopoulos 

Academic Editor

PLOS ONE